# Mechanisms, Effects, and Management of Neurological Complications of Post-Acute Sequelae of COVID-19 (NC-PASC)

**DOI:** 10.3390/biomedicines11020377

**Published:** 2023-01-27

**Authors:** Ian Z. Ong, Dennis L. Kolson, Matthew K. Schindler

**Affiliations:** 1Perelman School of Medicine, University of Pennsylvania, Philadelphia, PA 19104, USA; 2Department of Neurology, University of Pennsylvania, Philadelphia, PA 19104, USA

**Keywords:** COVID-19, SARS-CoV-2, PASC, long COVID, neurology

## Abstract

With a growing number of patients entering the recovery phase following infection with SARS-CoV-2, understanding the long-term neurological consequences of the disease is important to their care. The neurological complications of post-acute sequelae of SARS-CoV-2 infection (NC-PASC) represent a myriad of symptoms including headaches, brain fog, numbness/tingling, and other neurological symptoms that many people report long after their acute infection has resolved. Emerging reports are being published concerning COVID-19 and its chronic effects, yet limited knowledge of disease mechanisms has challenged therapeutic efforts. To address these issues, we review broadly the literature spanning 2020–2022 concerning the proposed mechanisms underlying NC-PASC, outline the long-term neurological sequelae associated with COVID-19, and discuss potential clinical interventions.

## 1. Introduction

COVID-19 is a disease caused by severe acute respiratory syndrome coronavirus 2 (SARS-CoV-2). As of 25 January 2023, there have been over 600 million confirmed cases of COVID-19 and over 6 million deaths worldwide [1]. Although the majority of people recover from acute COVID-19, an estimated 80% of people exhibit at least one symptom, sign, or laboratory parameter which persists two weeks after acute infection [2]. This presence of symptomatology following acute infection has been referred to in the medical community as “long COVID” [3], “chronic COVID syndrome” [4], “post COVID-19 syndrome” [5,6], and the more recent and widely-accepted term “post-acute sequelae of SARS-CoV-2 infection” (PASC) [7], while being referred to in the lay community as the “long-hauler” syndrome [4].

The symptomatology of PASC is highly variable as the condition can present with heterogeneous sequelae across multiple organ systems, and it can affect people independently of age and severity of acute infection [8,9]. Some sequelae may be a continuation of acute symptoms, or new symptoms may arise early in the recovery period. The hallmarks of PASC include generalized fatigue (58% of infected patients), exertional malaise, palpitations, and shortness of breath, and the most commonly reported neurological complications of PASC (NC-PASC) include headache (44%) and inattention (27%) [2,10]. Major limitations to these reports are the reliance on subjective self-reported symptoms and a lack of prospective observational clinical studies. Thus, NC-PASC remains poorly characterized.

## 2. Defining NC-PASC

The WHO defines the “post COVID-19 condition” as an illness that typically arises within three months of infection onset and causes symptoms and effects that last at least two months [11]. However, there is no consensus on the amount of time that must elapse until patients are considered to have PASC. For instance, recent literature has tried to define post-acute COVID-19 as symptoms and complications which present over 4 weeks after symptom onset [7,12,13]. This post-acute period may then be divided into a subacute period (4-12 weeks) and a chronic period (>12 weeks) [7]. However, it remains unclear as to the length of time between acute infection resolution and new symptomatology that one can invoke PASC as the likely diagnosis. In the case of NC-PASC, many patient-reported symptoms have been described, but few prospective objective studies have been published. In the following sections, we will characterize NC-PASC, a subset of PASC, by identifying common clinical features.

### 2.1. Clinical and Cognitive Features

Neurological symptoms are common in people experiencing PASC [14]. A meta-analysis of 81 subjective and objective symptom evaluation studies estimated that 22% of people report cognitive impairment 12 or more weeks following acute infection [15]. A cohort study using control data from the US Department of Veterans Health Care Systems showed that COVID-19 patients had an increased risk of stroke, cognitive impairment, memory problems, peripheral nerve disorders (including Guillain–Barré syndrome), and movement disorders (among other disorders) 30 days after an acute COVID-19 infection [16]. Other common self-reported neurological symptoms include cognitive impairment (frequently referred to as “brain fog” and which typically involve difficulty with memory, attention/concentration, and multitasking), new or worsening headaches, numbness/tingling, dysgeusia, anosmia, dizziness, blurry vision, and tinnitus [17].

The acute symptomotology of COVID-19, while not a focus of this review, warrants discussion as it can pertain to NC-PASC development. During an active infection, COVID-19 can cause encephalitis [18,19], meningitis [20,21], myelitis [22], demyelination [23], and seizures [24]. However, there is limited evidence that these acute entities predict long-haul symptoms as many people report neurological symptoms following mild COVID-19. Although, there have been reports of increased risk for seizures during COVID-19 recovery [25,26]. Treatments for acute COVID-19 can also cause neurological manifestations similar to NC-PASC; for example, a man with severe rheumatoid arthritis developed multifocal cerebral thrombotic microangiopathy following infusion with tocilizumab, an IL-6 receptor approved for the treatment of COVID-19 [27,28]. Acute brain injuries as caused by active infection or treatment are rare, but they should be considered in the holistic management of COVID-19 patients.

There are currently no diagnostic criteria for NC-PASC [29]. This may in part be due to a lack of consensus on the proper scope of “neurological” issues. For instance, Moghimi et al. consider neurological effects broadly, proposing a set of criteria for NC-PASC which necessitates multiple manifestations from at least some of the following categories: neurologic, neurocognitive, neuroendocrine, autonomic dysfunction, immune system, laboratory findings, reduction in the ability to engage in activities of daily living (ADL), fatigue, neuromuscular, and neuropsychiatric [10]. For the purposes of this review, we will narrow our scope and examine “NC-PASC” as sequelae primarily affecting the central nervous system (CNS).

### 2.2. Risk Factors Associated with the Development of NC-PASC

Examining the etiology of NC-PASC may assist clinicians in enacting prophylactic measures to slow or prevent the development of PASC. Unfortunately, the presentation of NC-PASC is extremely heterogeneous, and the factors that influence their development are not clearly known [30]. Considering PASC as a whole, a simple model applied to self-reported symptom data suggests that increasing age and female sex are predictive of sequelae which last over 28 days, whereas body mass index is predictive of sequelae that last between 10 and 28 days [31]. Also, patients who adhere to a diet rich in saturated fats, sugar, and refined carbohydrates (the “Western diet”) are especially susceptible to severe COVID-19 infection and its long-term effects [32].

With respect to NC-PASC, lingering damage and physiological perturbations caused by acute COVID-19 infection are recurring themes in proposed mechanisms of sequelae development. For instance, Doykov et al. observe a persistent inflammatory response 40–60 days post-infection in asymptomatic and mildly affected (not hospitalized) patients [33]; this sustained inflammation is a potential long-term pathway for sequelae development. NC-PASC may also be attributed to microcirculation damage including endothelial cell swelling, microthromboses, capillary blockage, and pericyte damage [34]. Together, inflammation and physiological stress caused by circulatory system disruptions and a heightened immune response could be instrumental in the development of NC-PASC. Patients who are more susceptible to these disease pathways are thus at greater risk for developing long-lasting neurological complications. Finally, certain comorbities during acute infection such as diabetes, chronic obstructive pulmonary disease (COPD), cardiovascular disease, hypertension, and human immunodeficiency virus (HIV) infection should be considered as a more severe COVID-19 infection in the acute setting may increase susceptibility to and severity of NC-PASC [35]. Notably, ischemic stroke predicts worse outcomes both as a prior event in a patient’s history and as a complication of acute COVID-19 [36,37].

## 3. Hypothesized Pathophysiology of NC-PASC

Understanding the pathophysiological mechanisms underlying NC-PASC could yield more effective approaches to treatment. However, discordance exists within the literature as to how SARS-CoV-2 infection causes lingering neurological symptoms; namely, do symptoms arise from direct effects of SARS-CoV-2 neuroinvasion and injury in the CNS, or indirect effects causing secondary downstream injury or neurodysfunction? We will consider these possibilities in turn. A graphical summary of the hypothesized pathophysiology of NC-PASC is provided in Figure 1.

### 3.1. Is SARS-CoV-2 Neurotropic?

One broad pathophysiological mechanism supported by a few studies and case reports is neuroinvasion (neurotropism), in which the virus invades the CNS [38,39,40,41,42,43]. This is distinct from neuronotropism, which refers to the capability of a virus to infect neurons in vivo. Known neuronotropic viruses including Zika, polio, mosquito-born viruses (arboviruses, including West Nile Virus), and herpesviruses can target and destroy neurons, leading to neurological sequelae [44,45,46,47]. Post-mortem examination and brain autopsies have suggested that SARS-CoV-2 neuroinvasion occurs, but only in a few acutely infected patients with severe clinical manifestations. Furthermore, there are only rare case reports of detection of SARS-CoV-2 in cerebrospinal fluid (CSF) also in severely symptomatic patients, and this is associated with encephalitis, stroke, and in one patient, Guillain-Barré syndrome [48,49,50].

The most rigorous and comprehensive studies of post-mortem brain tissues from infected patients have failed to provide strong evidence in support of the capability of SARS-CoV-2 to spread within the CNS, but provides evidence for the potential for initial neuroinvasion. Although the affinity of SARS-CoV-2 to the respiratory and olfactory epithelium points to olfactory sensory neurons as a potential route of neuroinvasion [51], samples of olfactory mucosa and olfactory bulb from 70 recently deceased COVID-19 patients (postmortem interval of approximately one hour) and 15 controls fail to show evidence of infection in olfactory mucosa or bulb neurons, suggesting that SARS-CoV-2 does not infect olfactory neurons in vivo [52]. Other studies, however, do demonstrate the presence of SARS-CoV-2 RNA sequences in several anatomically distinct and distant brain regions during the acute and sub-acute phases of infection [40,41,42]. In Solomon et al., several autopsied patients (<32 days from infection) had viral sequences detected (by PCR) at very low levels (5.0–59.4 RNA copies/mm^3^) in medulla, frontal lobes, and olfactory nerves, although viral proteins were not detected by immunohistochemical labeling [41]. There were no NC-PASC patients in that study. Similarly, in Meinhardt et al., PCR detection of SARS-CoV-2 RNA sequences in CNS tissue in one patient was observed at day 60 post-infection, and not thereafter [42].

Finally, Stein et al. demonstrated varying levels of SARS-CoV-2 RNA (genomic/sub-genomic) in 44 recently deceased COVID-19 patients throughout non-respiratory tissues, including the brain (Figure 2) [40]. The range of expression of SARS-CoV-2 RNA copies varied by approximately 7 logs, with a dramatic reduction in detectable RNA copies between days 4–30. The detection of extremely low copy numbers (10^−1^ to 10^−3^ copies/ng RNA) in several CNS regions (cervical cord, thalamus) at day 230 in one patient does indeed demonstrate some residual SARS-CoV-2 virus sequences in that patient. Although replicating SARS-CoV-2 was cultured from the thalamus from one patient at day 13 in the study, evidence for ongoing replication of the virus in the brain that could contribute to NC-PASC is lacking. Thus, while there is evidence that SARS-CoV-2 is able to invade the CNS in the severe, acute infection phase, there is extremely limited data to support the persistence of SARS-CoV-2 being an etiology for NC-PASC [43]. The SARS-CoV-2 RNA expression data collected by Stein et al. are adapted and presented in Figure 2.

Overall, these studies support the hypothesis that SARS-CoV-2 is neuroinvasive and that invasion into the CNS occurs, and that these observations are most consistent with hematogenous spread given the simultaneous appearance of SARS-CoV-2 RNA in anatomically distinct and distant brain regions. In contrast, trans-neuronal viral spread into the CNS would be supported by detection of SARS-CoV-2 RNA in contiguous and anatomically connected brain regions over time. Thus, although SARS-CoV-2 is neuroinvasive in some severely ill patients during acute infection, there is little, if any, evidence for a functional SARS-CoV-2 reservoir within the CNS after resolution of the acute infection. Therefore, the genesis of NC-PASC symptoms likely depends on secondary effects of acute infection.

There is some evidence to suggest that NC-PASC could arise from microcirculatory damage and/or immune system dysfunction, primarily by means of hypoxia, cerebrovascular disease, and cytokine-mediated neuroinflammation [15]. In the remaining sections, we will discuss the pathways by which individual neurological sequelae may arise from these broad mechanisms. Although other mechanisms adjacent to COVID-19 hospitalization such as medication effects and post-intensive care syndrome have been implicated in sequelae development [53], we will focus here on the two main pathways that are the most substantiated and central to the neuropathogenesis of post-acute COVID-19. Note that there is likely not a single pathway that explains all NC-PASC in every individual. A particular clinical phenotype may reflect components of each theorized mechanism working in combination; furthermore, the same observed phenotype may be driven by different mechanisms in different individuals.

### 3.2. Hypoxia-Associated NC-PASC

Many acutely ill COVID-19 patients present with some degree of hypoxia, even if they do not appear to be under respiratory stress [54]. Autopsy studies have shown acute hypoxic injury in the cerebrum and cerebellum with accompanying neuron loss in patients who died from complications from severe acute COVID-19 infection [41,55]. Also, acute hypoxia due to cardiac arrest and acute respiratory distress syndrome is associated with cognitive deficits which last long after recovery [56,57]. Cognitive impairment could theoretically arise from acute hypoxia experienced as a secondary effect of COVID-19-induced pneumonia.

A cross-sectional study of COVID-19 patients conducted by Dondaine et al. observed that those people who received oxygen therapy to manage pneumonia also had more severe memory impairment, attentional disorders, and psychomotor slowing months after initial recovery than people who did not require supplemental oxygen therapy for their acute COVID-19 infection [58]. Importantly, the authors noted no differences in fatigue, anxiety, or depression scores between the two groups, suggesting that hypoxia has no effect on these outcomes.

Endothelial damage resulting from hypoxia can contribute to cell death and cytokine signaling, upregulating an inflammatory response which potentially underlies the cognitive impairment reported by some patients [59]. However, studies of acute lung injury in porcine models have suggested that hypoxemia and an overactive inflammatory response contribute independently to brain damage [60,61]; they are thus distinct pathways of sequelae development. Hypoxemia may interact with systemic hypotension in a way that exacerbates brain injury and increases mortality in COVID-19 patients, but the details of this mechanism remain unclear [62,63,64]. Another proposed mechanism identifies hypoxia as a driver of viral replication and mitochondrial dysfunction, eventually driving NC-PASC by means of incorporation of the SARS-CoV-2 genome into the mitochondrial matrix [65,66]. Targeted research into the mechanisms of neurological damage as caused by COVID-19-mediated hypoxia is needed to substantiate these claims.

#### Cerebral Microbleeds

Cerebral microbleeds are small hemorrhages in the brain associated with structurally abnormal vasculature [67]. They are associated with increased mortality, increased illness severity, and poor functional outcomes following acute COVID-19 infection [68]. A study of 481 patients who received MRI scans on request places the prevalence of microbleeds in patients with COVID-19 at about 5% [69]. MRI imaging suggests that severely infected COVID-19 patients exhibit unusual patterns of microbleeds mainly affecting the corpus callosum [70,71]. Microbleeds with associated endotheliitis have also been observed in histopathological studies within the brainstem, cerebellum, neocortex, and gray and white matter in patients who died 5–15 days after admission to the hospital from complications of severe COVID-19 infection [72].

Although an association between COVID-19 and the development of cerebral microbleeds has been established, the underlying pathogenic mechanism is still under investigation. A retrospective case series conducted by Dixon et al. suggests that microbleed patterns in COVID-19 patients are similar to those seen in critically-ill non-COVID-19 patients and in patients with severe hypoxia from other causes [73]. Thus, the question remains as to whether microbleeds are a result of COVID-19-mediated endotheliitis [72] or are a generic feature of severe illness or hypoxia. Finally, the connection between microbleeds, other prominent NC-PASC such as cognitive impairments (“brain fog”), and COVID-19-associated coagulopathies should be considered [74]. One theory potentially linking microbleeds and cognitive dysfunction involves ferroptosis, an iron-mediated mechanism of cell death which may accelerate brain aging [75]. However, the low rate of observed microbleeds is not consistent with the prevalence of NC-PASC and therefore cerebral microbleeds cannot be the only mechanism leading to long-term neurological sequelae.

### 3.3. Immune-Associated NC-PASC

Chronic systemic inflammation as caused by a dysregulated immune system has been proposed as a pathway for the development and persistence of NC-PASC. Acutely, COVID-19 can cause severe inflammation via “cytokine storm” [76,77] and subsequent pyroptosis of immune cells including monocytes [78]. While this process is typically time-limited, persistent low-grade inflammation (LGI) has been hypothesized to drive NC-PASC through the continuous release of damage-associated molecular patterns (DAMPs) by tissues and cells damaged by COVID-19 [79]. LGI can lead to the dysregulation of brain microglia, resulting in heightened neuroinflammation and increased cytokine release [79]. The blood brain barrier (BBB), an important protective and supportive structure of the brain, is sensitive to an ongoing immune response [54]. In particular, BBB disruption has been linked with a number of long-term neuropathologies such as dementia and multiple sclerosis, and it has been speculated that SARS-CoV-2 infection could exacerbate these pathologies [57,80]. However, the mechanisms linking LGI with neurodysfunction and neurodegeneration remain unclear.

Lymphopenia in acute COVID-19 infection has been associated with disease severity and poor outcomes [81,82,83]. Patients presenting with lymphopenia show significant decreases in CD4+ and CD8+ T cells as well as in B cell counts; the former however is more closely related to a poor prognosis [84]. Jing et al. find that CD19, a crucial regulator of receptor signaling, is significantly reduced in the B cells of recovered patients [85]. While immunodeficiency has been linked with a variety of neurological deficits [86], the long-term effects of COVID-19-mediated lymphopenia on the brain have yet to be investigated in detail.

Several reports also suggest late complications can develop by means of autoimmune pathways. Two potential mechanisms include SARS-CoV-2 as a direct trigger of autoimmune events (potentially via molecular mimicry) or as an indirect trigger via systemic inflammation and immune dysregulation [87,88]. A study by Song et al. demonstrated CNS-specific T and B cell activation responses in cells isolated from the CSF of individuals with neurological symptoms in acute COVID-19 infection (days 2–43) [89]. Furthermore, CSF-derived immunoglobulin G (IgG) from 5 of 7 patients in the study showed reactivity against neurons. This study suggests that autoimmune responses (cellular and humoral) within the CNS compartment may contribute to the development of NC-PASC. Another autoimmune condition associated with the neurological system in COVID-19 patients is Guillain-Barré syndrome (a peripheral nerve demyelinating disease), although this is extremely rare in COVID-19 patients [90]. Other autoimmune driven diseases outside of the neurological system have been described, such as immune thrombocytopenic purpura, autoimmune hemolytic anemia, and Kawasaki disease [90]. Similar to the cerebral microvascular theory of NC-PASC, the frequency of objective findings consistent with autoimmunity is much lower than the frequency of NC-PASC. Further investigation is required to elucidate how immune dysregulation may lead to (and gauge the risk for) the development of NC-PASC, particularly in those with preexisting autoimmune conditions [88].

## 4. Comparison to Other “Long-Haul” Viral Diseases

Viruses other than SARS-CoV-2 are also known to trigger the development of post-viral syndromes. For instance, myalgic encephalomyelitis/chronic fatigue syndrome (ME/CFS) is a complex, multi-systemic, chronic condition characterized by profound neurological and immunological dysfunction following acute viral infection [91,92,93]. The etiology of ME/CFS remains unknown, but overlapping symptomatology with NC-PASC such as post-exertional malaise could potentially point to shared pathogenic mechanisms [93]. Thus, synthesizing information across multiple viral diseases may provide insight into the mechanisms of NC-PASC development and uncover new avenues for research. Here, we briefly review other viruses implicated in “long-haul” conditions and draw comparisons between methods of infection and neurological symptomatology. The information discussed is organized and presented in Table 1.

### 4.1. SARS-CoV-1

The SARS-CoV-1 virus operates analogously to SARS-CoV-2 in that they both target and bind to the mammalian ACE2 receptor using spike proteins [94]. SARS-CoV-1 infiltrates the epithelia of the respiratory tract via this mechanism and chiefly causes flu-like symptoms such as fever, chills, and cough [95]. The literature has suggested a common course for post-acute COVID-19 and SARS and thus a hypothetical shared post-coronavirus symptomatology. As noted by Patcai, a subset of SARS patients at their hospital developed “long SARS,” a non-specific, ill-defined condition bearing many similarities to PASC [96]. Neurological complications are rarely observed during follow-ups of SARS patients, but cases of musculoskeletal disorders and peripheral neuropathy have been described [97,98]. Furthermore, Guillain-Barré syndrome and ischemic stroke were also observed in a small number of patients following SARS-CoV-1 infection, echoing the earlier discussion of NC-PASC pathophysiology [94]. Overall, the scant reports of SARS-CoV-1-associated neurological sequelae contrast strikingly with the high proportion of COVID-19 patients who go on to experience NC-PASC [14].

**Table 1 biomedicines-11-00377-t001:** Overview of viruses associated with “long-haul” neurological symptoms. Primary infection targets and major neurological sequelae are listed.

Virus Name(s)	Family	Primary Targets	Major Neurological Sequelae
SARS-CoV-2	Coronaviridae	Ciliated and alveolar type II cells in airway and alveolar regions [99,100]	Headache, inattention [2,10], cognitive impairment [15], memory impairment, peripheral nerve disorders (including Guillain-Barré syndrome), movement disorders, stroke [16], numbness/tingling, dysgeusia, anosmia, dizziness, blurry vision, and tinnitus [17]
SARS-CoV-1	Coronaviridae	Respiratory tract epithelial cells [95]	Cognitive impairment, sleep disturbance [96], neuromusculoskeletal disorders [97], peripheral neuropathy [98], Guillain-Barré syndrome, and stroke [94]
HSV-1, HSV-2, VZV	Herpesvirinae	Epithelial cells, sensory ganglia during latency [101]	Herpes labialis (cold sores), herpes zoster (shingles), and encephalitis [102]
EBV	Herpesvirinae	B cells, nasopharyngeal epithelial cells, and CNS neuronal cells [103]	Combative behavior, seizures, headache [104], “Alice in Wonderland” syndrome, facial nerve palsy, progressive microcephaly, and encephalitic illness [105,106]
CMV	Herpesvirinae	Epithelial cells, endothelial cells, fibroblasts, smooth muscle cells [107], and CNS resident cells [108]	Sensorineural hearing loss, deficits in language function, [109] and learning disability [110]
WNV, JEV, TBEV, ZIKV	Flaviviridae	Dendritic cells and CNS cells [47,111]	Memory impairment, behavioral changes, speech disability, movement disorders, and seizures [47]
CHIKV	Togaviridae	Broad organ and cell tropism [112], stromal CNS cells [113]	Cognitive disturbance [47] and delayed cognitive development [114]
HIV-1	Retroviridae	CD4+ immune cells [115], CNS macrophages, microglia, and T lymphocytes [116]	Dementia, sensory neuropathies, and encephalitis [115,116]

### 4.2. Neuronotropic Viruses

Other viruses contribute to long-term symptoms by directly infecting neurons (neuronotropic viruses) and causing cellular damage. Although SARS-CoV-2 is likely not neuronotropic (not capable of infecting neurons in vivo), understanding the post-acute course of neuronotropic viruses could help guide diagnoses and better elucidate the relationship between neuronal dysfunction (NC-PASC) and direct neuronal infection. Alpha-herpesviruses such as varicella-zoster virus (VZV), herpes simplex virus 1 (HSV-1), and herpes simplex virus 2 (HSV-2) can establish latent neuronal infection in sensory ganglia following primary infection and cause neurological symptoms upon reactivation [101,117]. HSV reactivation is associated with cold sores and rarely encephalitis whereas VZV reactivation causes herpes zoster, a painful vesicular skin rash [102]. Epstein-Barr virus (EBV) can disrupt and cross the BBB, replicate within the CNS, and cause neuroinflammation [103]. EBV-associated BBB injury and neuronal damage can lead to neurocognitive impairment, which may be long-lasting. This may occur in cases of EBV-associated multiple sclerosis, encephalitis, meningitis, and encephalomyelitis [103]. The beta-herpesvirus cytomegalovirus (CMV) infects CNS resident cells, including neurons, astrocytes, and endothelial cells, which can result in a latent or active replication state of infection [108]. Congenital CMV infection causes progressive sensorineural hearing loss in 13–15% of affected newborns [110] and some deficits in language function later in life [109].

Although CNS infections caused by arboviruses (arthropod-borne viruses) are generally associated with acute CNS symptoms, some infections are also associated with long-term neurocognitive sequelae [47]. Among these viruses that are only recently being appreciated for such sequelae are West Nile virus (WNV), Japanese encephalitis virus (JEV), tick-borne encephalitis virus (TBEV), and Zika virus (ZIKV). Mechanisms of persistent cognitive impairment may include not only direct neuronal infection and injury and impairment of neuronal development (including adult neurogenesis), but also indirect effects mediated by the release of toxic or inflammatory factors by infected glia or immune cells. Virus-induced modification of host defensive response mRNAs is well-described [118], and one may speculate about host mRNA modifications within the CNS having an impact on cognitive functioning [47], although no such data have been published. Zika virus infection can lead to profound neurological complications including Guillain-Barré syndrome and inflammation of the brain and spinal cord [119]. In summary, neuronotropic viruses such as herpesviruses and certain arboviruses can cause long-term neurological symptoms similar to that seen in patients experiencing NC-PASC, possibly hinting at certain pathophysiological commonalities involving inflammation, BBB integrity, and peripheral sensory deficits.

### 4.3. Non-Neuronotropic Viruses

NC-PASC development may in part be driven by the secondary dysregulation of the immune system and response. However, other non-neuronotropic viruses more directly target immune system components, leading to lasting neurological effects. HIV-1 is a neuroinvasive/neurotropic retrovirus that infects CD4 positive immune cells (but not neurons), ultimately leading to systemic immune deficiency when left untreated [115]. Persistent immune activation in later stages of infection can cause chronic inflammation of the CNS and PNS, leading to neurological sequelae such as dementia and sensory neuropathies [115]. HIV-1 establishes a persistent infection within macrophages and microglia in the CNS [116]. Untreated HIV infection is associated with a 20% risk for development of HIV encephalitis and associated dementia, which results from neurodegeneration induced by active HIV replication within the brain and indirect mechanisms of neuronal cell injury [120]. In people living with HIV (PWH) who are on suppressive antiretroviral therapy (ART), replication in the brain and other body compartments is “suppressed” to undetectable levels. Nonetheless, the persistence of replication-competent HIV within brain cellular reservoirs is associated with a risk for worsening cognitive functioning (albeit not dementia) as the patient ages [120]. Acquired immunodeficiency syndrome (AIDS) may increase the likelihood of opportunistic infection with other viruses such as cytomegalovirus, leading to the compounding of neurological symptoms as discussed in the previous section [121].

EBV primarily infects B lymphocytes, but it can also infect nasopharyngeal epithelial cells and neurons [103]. Following the acute period, it can maintain a latent infection in long-lived memory B cells [122]. In addition to lymphoproliferative disorders upon reactivation, a case series of 10 pediatric patients also implicates EBV reactivation in the development of neurological complications such as “Alice in Wonderland” syndrome, facial nerve palsy, progressive microcephaly, and encephalitic illness [105,106].

Some non-neuronotropic viruses primarily target non-immune cellular components. For instance, chikungunya virus (CHIKV), a non-neuronotropic mosquito-borne virus, can infect a wide range of different cells and tissue types including stromal CNS cells [112,113]. CHIKV infection is associated with cognitive disturbance and delayed cognitive development in infants under six months old, echoing reports of “brain fog” in patients with NC-PASC [47,114]. Understanding how non-neuronotropic viruses interact with both the host immune system and supportive tissues could inform treatment options for immunocompromised patients with NC-PASC and elucidate general pathways for NC-PASC development.

## 5. Future Directions

Research efforts are a crucial part of improving how the neurological sequelae of COVID-19 infection are managed. Here, we identify broad gaps in knowledge as suggested by previous work in the field and discuss the current state of research with regards to treating NC-PASC.

### 5.1. Knowledge Gaps and Research Avenues

Additional experimentation is required to substantiate the proposed pathways of NC-PASC development and answer outstanding questions. First, the specific mechanisms that link acute hypoxia from COVID-19-associated pneumonia to NC-PASC, such as cognitive deficits, remains unclear. Rahman et al. suggest that a direct effect of the virus on the nervous system (among other systems including the HIF-1 pathway) may play a role in the development of a persistent “silent hypoxia” that can lead to severe tissue damage [59]. Neuroimaging studies involving COVID-19 patients may allow for the identification of specific regions damaged by hypoxia that correlate with NC-PASC [58,123]. The role of hypoxia and compromised mitochondrial function in neurons in the development of chronic brain fog is also worth exploration [65,66]. A more thorough understanding of mitochondrial disruption in COVID-19 on the genomic level could inspire mitochondria-based therapeutics for the management of hypoxia-related NC-PASC. Finally, many questions remain surrounding the temporal evolution of COVID-19 associated microbleeds and how abnormal vascular events mechanistically result in NC-PASC [72]; longitudinal imaging studies could shed light on the neurological effects and evolution of cerebral microbleeds.

Dysregulation of the immune system is a central feature of poor outcomes of COVID-19 infection. Future studies could further investigate the effect of the virus on BBB integrity and characterize how a pro-inflammatory state disrupts this barrier on the cellular level [80]. Certain cytokines such as interleukin 1 beta have been observed to damage the hippocampus and impede learning and recall in mouse models with septic encephalopathy, suggesting a still unclear role of the “cytokine storm” in brain damage and the development of lasting neurological symptoms [57,124]. Studies could also aim to untangle the relationship between metabolic disruptions and chronic B cell lymphopenia and to determine which factor is more important in NC-PASC development [85]. Finally, more work is needed to elucidate how SARS-CoV-2 can trigger autoimmune responses and how these aberrant responses can lead to long-term neuropathology including Guillain-Barré syndrome [88]. Quantifying the extent to which COVID-19 contributes to the many pathways discussed above and determining how activation of these pathways causes clinical symptoms are the largest hurdles for better understanding the pathophysiology of NC-PASC.

#### Methodological Recommendations

Heterogeneity in studies of NC-PASC can impede meta-analyses and may present as differences in methods of symptom assessment and data collection [15]. To improve data quality, screening tools used for assessing NC-PASC should include more sensitive tests such as the Screen for Cognitive Impairment for Psychiatry (SCIP) and the THINC-integrated tool (THINC-it) [15]. Collaboration and the pooling of patient data across institutions could also increase the power of statistical analyses and allow for more generalizable results.

As the long-term effects of COVID-19 infection become increasingly prevalent in the coming years, longitudinal prospective studies could be key to furthering our understanding of NC-PASC. Such studies could follow cohorts of patients and note exposures that may increase the risk of developing certain NC-PASC such as fatigue or cognitive impairment. Neuroimaging studies could also be leveraged to identify significant relationships between structural damage to brain tissue and vasculature and chronic symptomatology.

Finally, there are no criteria for the diagnosis of NC-PASC excepting a proposed rubric by Moghimi et al. [10]. This is due in part to the lack of consensus as to the time at which ongoing symptoms can be considered PASC, contention over the proper scope of “neurological” issues, and uncertainty as to whether certain PASC are neurological in origin. As novel treatments for PASC emerge, treatment timelines and therapeutic targets could inform a definition which best integrates with clinical practice and promotes good outcomes. A standard definition for NC-PASC could also streamline research efforts, ultimately resulting in more focused analyses and useful results.

### 5.2. Treatment Approaches

At present, no drug has been shown to specifically and effectively reduce symptoms of PASC, including neurological complications [8,53,125]. Thus, formulating management strategies for managing NC-PASC necessitate a holistic yet pragmatic perspective [126]. For instance, one treatment regimen could include psychological elements such as cognitive therapy [10] that work alongside ad hoc medication and targeted neurorehabilitation [53]. Further research is needed to determine whether treatments for ME/CFS such as antioxidant therapies are also effective for fatigue symptoms in long-haul COVID-19 patients [127,128].

Hyperbaric oxygen therapy (HBOT) has been reported to improve multiple neurological symptoms including fatigue, executive dysfunction, and inattention in patients experiencing NC-PASC [129]. A randomized controlled trial conducted by Zilberman-Itskovich et al. shows significant improvements in cerebral blood flow and microstructural changes in 37 NC-PASC patients who received HBOT [130]. Further, these changes were coupled with improvements in executive function, attention, fatigue (all associated with increased perfusion in Brodmann Areas 6, 8, 10), psychiatric symptoms (microstructure changes in the superior corona radiata area), and diffuse muscle and joint pain (increased perfusion in the insula, hippocampus, putamen, prefrontal cortex, and cingulate cortex) [130]. Despite these promising results, the generalizability of HBOT as a treatment for NC-PASC may be an issue owing to the relatively small size of these experimental studies. The development of novel therapeutics for post-acute symptoms will become increasingly important as the social and economic burdens of NC-PASC continue to compound.

NC-PASC may be prevented altogether by the proper management of acute COVID-19 infection. Adhering to current standards of care in patients with severe COVID-19 infection and resultant acute respiratory distress syndrome (ARDS) is important for reducing neurological damage that could have long-lasting effects [131]. Furthermore, mathematical models demonstrate the efficacy of vaccination as prophylactic treatment for COVID-19; promoting equitable access to the vaccine worldwide could reduce the overall incidence of COVID-19 and by extension NC-PASC [132].

Sphingosine has been shown to prevent the interaction of the SARS-CoV-2 spike protein with ACE2 [133], and thus sphingosine signaling pathways may be disrupted in COVID-19 patients. Given this, targeting the SphK-S1P-SIPR pathway could hold promise as an adjunctive therapy in patients with acute COVID-19. However, to the authors’ knowledge, there is no evidence that S1P receptor modulators are effective for the treatment of COVID-19. A case series of patients treated with fingolimod or siponimod (S1P receptor modulators) found that the risk of severe COVID-19 infection for the general population and those receiving S1P receptor modulators was similar [134]. Further, there are no data to support or refute the claim that S1P receptor modulators are effective in the prophylaxis of NC-PASC, although it has been speculated that the neuroprotective and neuroregulatory properties of S1P are disrupted in COVID-19 infection, which could be a driver of long-term sequelae [135]. Further research into the role of the SphK-S1P-SIPR pathway during infection is warranted.

As of 25 January 2023, clinical trials related to NC-PASC include observational studies aimed at characterizing the natural history of neurological sequelae [136,137] and a host of therapeutic trials exploring treatments such as immunotherapy [138,139] and nerve and tissue electrostimulation [140,141,142]. Overall, the trials included in this initial search on clinicaltrials.gov appear heterogeneous in design and intervention and thus show promise for tackling NC-PASC management from a number of different angles. Although studies aimed at characterizing the broad spectrum of involvements and neurological manifestations associated with NC-PASC are foundational, there is a clear and emerging need for trials of novel treatments as the burden of NC-PASC in the global population continues to increase.

## 6. Conclusions

The management of the neurological complications of post-acute sequelae of COVID-19 infection continues to be a concern in clinics worldwide. Post-mortem studies of brain tissue and observational studies identify pathways associated with hypoxia and dysregulation of the immune system as the main drivers of NC-PASC. However, due to a lack of experimental controls, the precise cellular mechanisms underlying these pathways remain unclear. Future research should focus on elucidating these mechanisms and leveraging this knowledge to advance novel therapeutics.

## Figures and Tables

**Figure 1 biomedicines-11-00377-f001:**
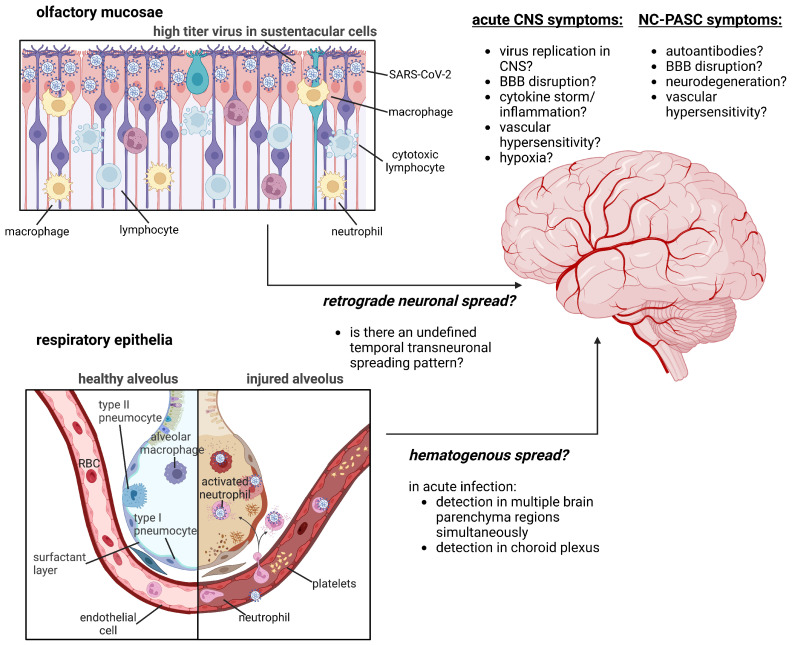
Summary of hypothesized pathophysiological mechanisms of NC-PASC development. Outstanding inquiries in the field concern the mechanism of SARS-CoV-2 neuroinvasion (direct neuronal infection via transneuronal spread through the olfactory mucosae versus hematogenous spread through transendothelial migration of infected leukocytes into the CNS) and the biochemical pathways that drive NC-PASC development and resolution. Figure was created with BioRender.com.

**Figure 2 biomedicines-11-00377-f002:**
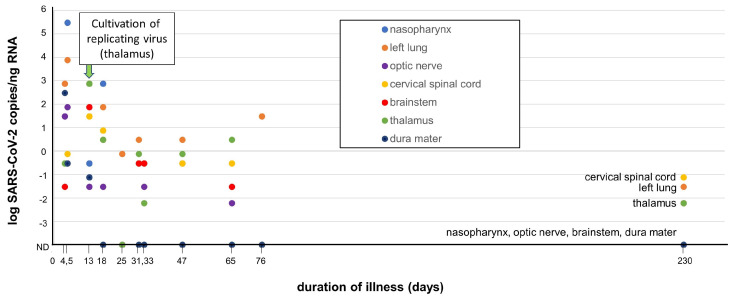
Summary of detection of SARS-CoV-2 in several autopsied tissues, including brain and nasopharynx (adapted from Stein et al. with permission [40]). SARS-CoV-2 RNA sequences were detected by qRT-PCR in autopsied tissues derived from SARS-CoV-2-infected decedents (n = 44) over a time range of 4–230 days. RNA levels are expressed as the log SARS-CoV-2 RNA copies/ng of total RNA vs. duration of infection at time of patient death. Isolation of replicating SARS-CoV-2 from the thalamus (day 13, arrow) was successful through co-culturing of the tissue in vitro with a susceptible Vero cell line. Several other non-brain tissues (respiratory tract, heart, lymph nodes, gastrointestinal tract, adrenal gland, and eye) also yielded SARS-CoV-2 replicating virus by co-culture during early infection (<14 days, not depicted in figure). Low levels of SARS-CoV-2 RNA was detected at day 230 in cervical spinal cord, left lung and thalamus, but not in nasopharynx, optic nerve, brainstem, or dura mater. Adapted with permission from D. Chertow [40].

## Data Availability

Not applicable.

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
