# Peer review of "Mechanisms, Effects, and Management of Neurological Complications of Post-Acute Sequelae of COVID-19 (NC-PASC)"

_biomedicines, 2023, doi:10.3390/biomedicines11020377_

Round 1
Reviewer 1 Report
The authors have chosen a very interesting and hot topic. This is a very nice literature review, and even if this is not a systematic review, it contains all the important information about the post-COVID neurological sequelae.
Only some minor comments.
Minor Comments:
Page 2, Line 62: Please expand the abbreviation ADL.
Page 2, Lines 69-72: Please make a comment on the severity of the disease that is in line with these factors (increasing age, high BMI etc.)
Page 3, Neurotropism: Please be careful when stating that SARS-CoV-2 is not neurotropic. Brain biopsies by Solomon et al, found RNA of the Virus to be in areas very remote to the olfactory mucosa, such as the cerebellum. This is hard evidence that there may be neurotropism. Even the microcirculatory hypothesis is in accordance with neurotropism. Please rephrase the paragraph and do not mostly support the non-neurotropic thesis.
Page 7, Lines 312-315: Please add reference doi: 10.1016/j.bbadis.2022.
Very nice manuscript overall.
Reviewer 2 Report
In this review paper the authors discussed proposed mechanisms underlying neurological complications of post-acute sequelae of SARS-CoV-2 infection (NC-PASC) and potential clinical interventions. This topic is important. Some concerns and suggestions are listed as below:
Regarding hypothesized pathophysiology of NC-PASC, these can be summarized using a figure.
SARS-CoV has the ability to reach the brain through the olfactory bulb and then causes transneuronal spread to the brain. The SphK-S1P-SIPR pathway plays an important role in invasion of SARS-CoV-2 infection, particularly in the central nervous system (Role of the SphK-S1P-S1PRs pathway in invasion of the nervous system by SARS-CoV-2 infection, Clin Exp Pharmacol Physiol. 2021). This point should be discussed.
One recent study showed that SARS-CoV-2 is widely distributed, predominantly among patients who died with severe COVID-19, and that virus replication is present in multiple respiratory and non-respiratory tissues, including the brain, early in infection. The authors detected persistent SARS-CoV-2 RNA in multiple anatomic sites, including throughout the brain, as late as 230 days following symptom onset in one case (SARS-CoV-2 infection and persistence in the human body and brain at autopsy, Nature, 2022).
In the whole review paper, the authors only mentioned cerebral microbleeds and immune-associated NC-PASC regarding neurological complications of post-acute sequelae of SARS-CoV-2 infection. However, other conditions following SARS-CoV-2 infection (such as encephalitis, meningitis, myelitis, demyelinating disorders, seizures, polyneuropathy) should not be ignored. This is a major concern.
Since the early onset of the pandemic, several treatments have been tested and developed to treat COVID-19. These treatments have reduced the severity of the disease and the time patients spend at the hospital. However, many of those treatments that are being used for COVID-19 have the potential to cause neurological symptoms.
The authors said that the hallmarks of PASC include generalized fatigue, exertional malaise, palpitations, and shortness of breath, and the most commonly reported neurological complications of PASC (NC-PASC) include fatigue (58% of infected patients), headache (44%), and inattention (27%). Some may argue that these are non-specific neurological complications. What do you think?
Reviewer 3 Report
We read with great interest the review by long et al titled: Mechanisms, Effects, and Management of Neurological Complications of Post-acute Sequelae of COVID-19 (NC-PASC). In fact, this area is highly timely and important to investigate,
the review provided a synopsis of the proposed mechanisms however one major weakness is the lack of depth in discussing the proposed different pathological pathways which I think may need some tables to illustrate the studies and how they were congruent in reaching each of the different mechanistic drivers of pathophysiology. *(Tables are needed)
the work has two major sections:
1-Hypothesized Pathophysiology of NC-PASC
2-Comparison to Other “Long-Haul” Viral Diseases
these sections would need some schematics that can summarize the different components (Schematics are needed)
the sections related to clinical trials and management are lightly discussed without death and may require more discussion or alternatively can e removed especially since the authors have provided Knowledge Gaps and Research Avenues and a conclusion to conclude the work.
Finally, there are major and extremely important areas that the authors should discuss in the section on Risk factors which is comorbid with other conditions such as stroke, brain injury, and cardiovascular diseases.
Important Recent references are not included and are excellent resources for the authors:
Stroke in SARS-CoV-2 infection: a pictorial overview of the pathoetiology
S Aghayari Sheikh Neshin, S Shahjouei, E Koza, I Friedenberg, ...
Frontiers in Cardiovascular Medicine 8, 649922
SARS-CoV-2 involvement in central nervous system tissue damage
MA Haidar, Z Shakkour, MA Reslan, N Al-Haj, P Chamoun, K Habashy, ...
Neural Regeneration Research 17 (6), 122
Neuroimmune disorders in COVID-19. <https://pubmed.ncbi.nlm.nih.gov/35353232/> Ariño H, Heartshorne R, Michael BD, Nicholson TR, Vincent A, Pollak TA, Vogrig A.J Neurol. 2022 Jun;269(6):2827-2839
Neurological manifestations of COVID-19: A comprehensive literature review and discussion of mechanisms. <https://pubmed.ncbi.nlm.nih.gov/34304141/> Johansson A, Mohamed MS, Moulin TC, Schiöth HB.J Neuroimmunol. 2021
Round 2
Reviewer 2 Report
The authors have addressed my concerns.
Reviewer 3 Report
excellent work